**Data Availability Statement:** All relevant data are within the paper and its Supporting information files.

**Funding:** This study was financially supported by The Norwegian Regional Health Trusts in the form

# Endothelial dysfunction in ME/CFS patients

**Miriam Kristine Sandvik**[1¤]*, **Kari Sørland**[2], **Elisabeth Leirgul**[3], **Ingrid Gurvin Rekeland**[2,4], **Christina Særsten Stavland**[4], **Olav Mella**[2,4], **Øystein Fluge**[2,4]

**1** Department of Psychiatry and Addiction, Telemark Hospital Trust, Skien, Norway, **2** Department of Oncology and Medical Physics, Haukeland University Hospital, Bergen, Norway, **3** Department of Heart Disease, Haukeland University Hospital, Bergen, Norway, **4** Department of Clinical Sciences, University of Bergen, Bergen, Norway

¤ Current address: The Norwegian Medical Association, Oslo, Norway.
* miriamsandvik@gmail.com

## Abstract

### Objective

A few earlier studies have found impaired endothelial function in patients with Myalgic Encephalomyelitis/Chronic Fatigue Syndrome (ME/CFS). The present study investigated large-vessel and small-vessel endothelial function in patients with ME/CFS.

### Study design

The study was a substudy of the RituxME trial, a national, multicenter, randomized, double-blind, placebo-controlled phase III study on the effect of rituximab vs. placebo in ME/CFS patients in Norway. Flow-mediated dilation (FMD) and post-occlusive reactive hyperemia (PORH) was measured at baseline and after 18 months of treatment in 39 patients and compared with healthy controls. Other outcome measures were symptom severity and various physical function measures.

### Results

ME/CFS patients had markedly reduced FMD compared to healthy controls at baseline (5.1% vs. 8.2%, p< 0.0001, adjusted for arterial diameter and sex), and significantly lower microvascular regulation measured by PORH than healthy controls (1354 PU vs. 2208 PU, p = 0.002). There were no differences between the treatment and placebo groups in symptom changes or vascular measures. As a group, the ME/CSF patients experienced a slight, but significant improvement in clinical symptoms after 18 months. PORH, but not FMD, was similarly improved (1360 to 1834 PU, p = 0.028). There was no significant correlation between FMD and PORH. There were non-significant tendencies towards associations between symptom severity/physical function measures and lower FMD and PORH, and a significant correlation between PORH and steps per 24 hours at baseline.

of a grant to ØF and OM. This study was also financially supported by The Norwegian Research Council in the form of a RituxME study grant (229035/H10) awarded to ØF and OM. This study was also financially supported by The Norwegian Ministry of Health and Care Services in the form of a grant to ØF and OM. This study was also financially supported by Kavli Trust in the form of a grant to ØF and OM. This study was also financially supported by MEandYou Foundation in the form of a grant to ØF and OM. This study was also financially supported by The Norwegian ME association in the form of a grant to ØF and OM. The funders had no role in study design, data collection and analysis, decision to publish, or preparation of the manuscript.

**Competing interests:** The authors have declared that no competing interests exist.

## Conclusions

ME/CFS patients had reduced macro- and microvascular endothelial function, indicating that vascular homeostasis may play a role in the clinical presentation of this disease.

## Introduction

Myalgic Encephalomyelitis/Chronic Fatigue Syndrome (ME/CFS) is a disease with a massive negative impact on quality of life for affected patients [1], with unknown etiology and no established effective treatment. Defined by the Canada consensus criteria [2], it affects 0.1–0.2% of the population [3]. The main symptoms are pronounced fatigue and post-exertional malaise, sleep dysfunction with lack of restitution, cognitive disturbances, pain and sensory hypersensitivity, and other symptoms related to neuroendocrine, autonomic, and immune function.

In previous studies, results indicated that a subgroup of ME/CFS patients experienced improvement of symptoms after therapeutic B-lymphocyte depletion using the monoclonal anti-CD20 antibody Rituximab [4–6]. Taken together with studies showing that elderly ME/CFS patients had an increased risk of B-cell lymphomas [7], and studies showing autoantibodies associated with the disease [8–10], these findings gave rise to a hypothesis that ME/CFS in a subgroup of patients may be caused by a B-lymphocyte-driven post-infectious immune dysregulation.

In order to explore this hypothesis further in a larger study, we conducted a randomized, double-blind placebo-controlled study (RituxME trial) [11]. The study did not report any significant differences in outcome measures between the rituximab and placebo groups, thus refuting the hypothesis that B-cell depletion could be beneficial for ME/CFS patients selected on Canadian consensus criteria.

The clinical presentation of ME/CFS, with major symptoms arising from several organ systems, indicates disturbance of a biological system with widespread functions. Some of the most common and distressing symptoms of ME/CFS, i.e. orthostatic intolerance and post-exertional malaise [2], and findings of reduced anaerobic threshold and lactate accumulation in tissues after exercise [12, 13] could be indicative of disturbances in the vascular homeostasis.

One previous study by Newton et al., investigating flow-mediated dilation (FMD) for large vessel endothelial function and post-occlusive reactive hyperemia (PORH) for microvascular function in ME/CFS patients, concluded that ME/CFS patients have both large and small vessel endothelial dysfunction as compared to age- and sex-matched controls [14]. FMD, when performed under standardized conditions, is a test mainly reflecting nitric oxide (NO) synthesis in endothelial cells [15]. Recently, another study by Scherbakov et al. found that peripheral endothelial dysfunction was frequent in ME/CFS patients and was associated with disease severity [16]. We measured FMD in 12 ME/CFS patients in 2014 and found that 10 out of 12 patients had low FMD values (in 5 out of 12 < 2%, median value 2.8%) compared to previous studies on healthy subjects in appropriate age groups [17, 18]. All ME/CFS patients had normal dilation responses after administration of sublingual glyceryl nitrate. This led us to suspect that reduced availability of endothelial-cell derived NO, with subsequent inadequate autoregulation of blood flow according to the metabolic demand of tissues, could be a symptom-causing mechanism for ME/CFS patients. To investigate both large-vessel- and small-vessel function, we therefore conducted substudies of two drug intervention trials for patients with ME/CFS at Haukeland University Hospital: a phase II trial of cyclophosphamide [19] and the

RituxME trial. The results of the cyclophosphamide substudy were recently published and showed that patients with ME/CFS indeed had reduced endothelial function affecting both large and small vessels compared to healthy controls. Changes in endothelial function did not follow clinical responses during follow-up [20].

Our hypotheses for the present substudy were 1) FMD and PORH would be reduced in ME/CFS patients, compared to the healthy controls, and 2) There could be a correlation between ME/CFS disease severity and FMD/PORH levels, and lastly 3) If the treatment group showed symptom alleviation, this could be accompanied by an improvement of endothelial function.

## Materials and methods

### Study design

The present study was a substudy of the RituxME trial [11]. RituxME was a national, multicenter, randomized, double-blind, placebo-controlled phase III study in Norway, with the Department of Oncology at Haukeland University Hospital, Bergen, as the main study center. The patients were treated with either rituximab or saline water, two infusions with two weeks' interval (500 mg/m², max 1000 mg), followed by maintenance infusions after three, six, nine and twelve months (500 mg, fixed dose). The intervention code was opened after 24 months of follow-up (ClinicalTrials.gov: NCT02229942; EudraCT: 2014-000795-25). The study was approved by the Regional Committees for Medical and Health Research Ethics in Norway (2014/365). The study design, patient inclusion and exclusion, randomization and interventions are thoroughly described in the main study article [11]. The 40 patients included in Bergen were invited to a substudy investigating endothelial function by FMD and PORH. Thirty healthy controls were included for PORH and iontophoresis analyses. These controls were similar to the patient group in gender distribution and age. To compare FMD measures, we included a control group of 66 healthy women of similar age who served as a control group in a different study, performed by the same physicians [17]. All patients and the healthy PORH controls gave their written informed consent to participate in the substudy. The consent process is described in the study protocol (S1 File). The FMD controls gave written, informed consent to participate in the original study. Anonymized FMD data were used in comparison to patient FMD data in the current study, which after consultation with the local ethics committee was considered to be within the realm of the original consent.

### Variables

The main outcomes were measures of endothelial function (FMD and PORH) at baseline and after 18 months. Measures of disease severity at baseline and symptom change/clinical response at 18 months were other important variables.

**Measures of symptoms and disease severity.** The severity of ME/CFS was categorized into five groups (mild, mild-moderate, moderate, moderate/severe, severe). The categorization into different severity groups was based on thorough clinical assessment by the physician at the time of inclusion, and supported by patient-reported function level and questionnaires. Self-reported questionnaires at baseline included the Short-Form-36 Health Survey (SF-36) (version 1.2) [21, 22] and the Fatigue Severity Scale [23]. The patients were equipped with an electronic SenseWear armband in the home setting for five to seven days, in order to record baseline number of steps per 24 hours as a measure of level of physical activity [24, 25]. SF-36 forms were completed every three months and the Fatigue Severity Scale every six months. Physical activity registration with SenseWear armband was repeated after 18 (17–21) months.

**Vascular analyses.** Vascular analyses were performed in the patients enrolled at the main study center in Bergen. Thirty-nine patients were included in the baseline analyses. Among these, three patients did not undergo vascular examinations at 18 months follow-up due to symptom severity in two patients and one trial drop-out.

*Flow-Mediated Dilation (FMD).* The assessment of endothelial function by flow-mediated vasodilation (FMD) was performed by experienced physicians (MKS, EL) in all patients, according to standards given by the International Brachial Artery Reactivity Task Force [26]. All participants were overnight fasting, had abstained from high-fat foods 24 hours before the examination, and from smoking and intake of C-vitamins, caffeine, and medication on the examination day. Vascular measurements were performed using the GE Vingmed (GE Vingmed, Vivid E9, GE, Horten, Norway) system, with a multifrequency linear probe, 6–13 MHz (M12L). Measurements of the brachial artery diameter were performed at rest, after 5 min occlusion of blood flow with the cuff on the forearm distal to the site of measurements, and finally after glyceryl nitrate spray 0.4 mg sublingually. Images were recorded in end-diastole. Post-occlusion measurements of the artery were performed at the point of maximum dilation. A more detailed description of the procedure can be found in a previously published paper [17]. Both physicians analyzed 10 randomly selected FMD measurements, and the inter-observer variability expressed as two-way random intraclass correlation (ICC (2,2)) was 0.99 for baseline measurements, both pre- and post-occlusive, and 0.78 for FMD measurements.

*Post-Occlusive Reactive Hyperemia (PORH).* The participants were prepared as described for FMD. The measurements of cutaneous perfusion were performed using a Periflux 5000 unit with laser Doppler technology (Perimed, Stockholm, Sweden). A laser Doppler probe was preheated to 32 degrees Celsius and positioned on clean, intact forearm skin on an area with maximum baseline perfusion of 10 Perfusion Units (PU). Microvascular perfusion of the skin was recorded for a minimum of five minutes before and two minutes after occlusion. The hyperemia response was expressed in Perfusion Units as the area under the curve during the first two minutes post-occlusion, minus the area under the curve during two minutes baseline.

*Iontophoretic application of acetylcholine.* At baseline, we also assessed the acetylcholine-induced changes in cutaneous perfusion in patients and controls. Application of the vasoactive substance acetylcholine generates endothelium-dependent vasodilation which may be detected by laser Doppler flowmetry using the Periflux 5000 unit (as above) [27]. Iontophoresis of acetylcholine measures microvascular reactivity, but not specifically skin endothelial function [27]. In this study, acetylcholine 10 mg/ml, 0.18 ml (Miochol-E powder for intraocular solution dissolved in sterile water) was applied to the skin of the left forearm using iontophoresis, i.e. the application of a small electric current of 20 microampere/min. The hyperemia response was expressed in Perfusion Units as area under the curve for 10 minutes. Due to the baseline results with wide variability in both patient and control group and no significant differences between the two groups, we chose not to repeat iontophoresis of acetylcholine at 18 months.

## Statistical analyses

SPSS versions 25 and 27 were used for statistical analyses. Chi square tests were used to compare groups for categorical variables and t-tests (equal variances not assumed) for comparing means of continuous variables. Normality was tested using the Shapiro-Wilk´s test. FMD in ME/CFS patients and in healthy controls were normally distributed, and differences in mean between groups at baseline were compared by analyses of covariance (ANCOVA) in order to adjust for sex and baseline arterial diameter. For PORH, data from the ME/CFS group showed normal distribution, while data were not normally distributed in the healthy control group,

and Mann-Whitney test was used to assess difference between groups. FMD and PORH by ME/CFS severity groups were tested by ANOVA and Kruskal-Wallis tests, respectively.

The variables that describe patient function (SF-36-PF, FSS, self-reported function level and mean steps per 26h) were not normally distributed, therefore, Wilcoxon signed rank tests for related samples were used to compare means at baseline and 18 months. Three patients, for whom we only had baseline vascular values, were excluded from these analyses. General Linear Models (GLM) for repeated measures were used to assess differences in changes of the outcomes measures (FMD or PORH), from baseline to 18 months, between the randomization groups (rituximab and placebo). The analysis syntax for GLM repeated measures for FMD and PORH, with treatment group (rituximab or placebo) as between-subjects factor, is included as a supporting file (S2 File). Spearman correlation analyses were used to investigate the correlation between FMD and PORH. For correlation analyses between vascular measures and measures of disease severity and function levels, categories of disease severity were pooled into three groups, to increase group sizes. Figures were made using SPSS 25 and GraphPad Prism ver. 8.

## Results

### Baseline characteristics

Baseline characteristics of the participants are reported in Table 1. Mean age was 35.9 years (SD 9.2), and 82% were women. Using five categories for disease severity, the patients were evenly distributed between the rituximab and placebo groups. The majority of patients had been ill for 5 to 10 years (59%). Comparisons of baseline characteristics for ME/CFS patients vs. control groups are reported in Table 2. The control group for PORH analyses was similar to the patient group in age and sex distribution. The participants in the healthy control group for FMD were all women, and slightly older. ME/CFS patients had significantly higher systolic and diastolic blood pressure than control groups, but within normal range (Table 2).

**Vascular measures—baseline.** Among ME/CFS patients, mean diameter pre-occlusion was 3.00 mm (SD 0.50 mm); mean post-occlusion diameter at maximum dilation was 3.15 mm (SD 0.51 mm). For healthy women, mean diameter pre-occlusion was 2.94 mm (SD 0.42 mm); mean post-occlusion diameter at maximum dilation was 3.19 mm (SD 0.44 mm). Compared by unpaired t-test (equal variances not assumed), mean flow-mediated dilation (FMD) in ME/CFS patients at baseline was 4.9% (SD 3.6), significantly lower than in healthy women (mean 4,9% vs 8.3%, p< 0.0001). This is also illustrated in Fig 1, panel A. Adjusted for sex and baseline arterial diameter in analyses of covariance (ANCOVA), mean FMD was still significantly lower in ME/CFS patients than in healthy controls, as shown in Table 2. In a sensitivity analysis, we excluded all men (N = 7) from the patient cohort, with marginal changes in the results (mean FMD estimated to 5.2% in ME/CFS patients vs 8.3% in healthy controls, p< 0.0001, adjusted for baseline arterial diameter).

After administration of glyceryl nitrate sublingually, mean diameter was 3.69 mm (SD 0.59) in ME/CFS patients, and mean dilation in per cent was 23.3%, similar to healthy women (mean diameter 3.60 mm (SD 0.47) and mean dilation 22.6%). Adjusted for sex and baseline arterial diameter in ANCOVA, the results were similar and not significantly different between the two groups (p = 0.79) (Table 2).

In ME/CFS patients, mean PORH at baseline was 1354 PU (SD 754 PU). This was significantly lower than the control group (2208 PU, SD 1363 PU, p = 0.004 analyzed by Mann-Whitney test (Table 2), and is also illustrated in Fig 1, panel B. There was no significant correlation between FMD and PORH in ME/CFS patients at baseline (Spearman correlation coefficient 0.16, p = 0.32) (Fig 1, panel C). Mean value (area under the curve) for acetylcholine-induced

**Table 1. Baseline characteristics.**

| All ME/CFS patients, baseline | N | % or mean (SD) |
|---|---|---|
| Age (years) | 39 | 35.9 (9.2) |
| Female sex | 39 | 82.1% |
| BMI (kg/m$^2$) | 39 | 23.5 (3.1) |
| Blood pressure systolic at rest, baseline (mmHg) | 39 | 123.0 (13.8) |
| Blood pressure diastolic at rest, baseline (mmHg) | 39 | 74.6 (10.8) |
| Heart rate at rest, baseline | 39 | 70.5 (12.8) |
| Disease severity (1–5) | 39 | |
| Mild | 8 | 20.5% |
| Mild-moderate | 9 | 23.1% |
| Moderate | 8 | 20.5% |
| Moderate-severe | 5 | 12.8% |
| Severe | 9 | 23.1% |
| Disease duration (years) | 39 | |
| 2–5 | 8 | 20.5% |
| 5–10 | 23 | 59.0% |
| 10–15 | 8 | 20.5% |
| SF-36-PF[1] | 39 | 31.2 (21.7) |
| Fatigue Severity Scale[2] | 39 | 59.7 (3.7) |
| Self-reported function level[3] | 39 | 18.0 (9.1) |
| Mean steps per 24h[4] | 39 | 3055 (2202) |
| FMD baseline (%) | 39 | 4.9 (3.6) |
| PORH baseline (Perfusion Units) | 39 | 1354 (754) |

[1]SF-36-PF = Short-Form-36 Health Survey Physical Function subscale. Raw score: range 0–100, higher score denote better function.

[2]Fatigue Severity Scale: range 7–63, higher scores denote worse symptoms.

[3]Self-reported function level, range 0–100%, according to a form with examples.

[4]Measured by Sensewear armbands, continuously for 5–7 consecutive days.

Abbreviations: FMD: Flow-mediated dilation; PORH: Post-occlusive reactive hyperemia.

hyperemia was 28,691 PU (SD 15,151) for the ME/CFS group and 28,557 PU (SD 15,418) for the control group, with no significant difference (p = 0.97).

**Associations between vascular measures and disease severity.** Correlation analyses between vascular measures and measures of disease severity and function levels are illustrated in Fig 2. There was a near-significant association between FMD at baseline and disease severity assessed as mild/moderate versus moderate versus moderate/severe (p for trend = 0.051, Fig 2, panel A), but no significant correlations between FMD and either SF-36 Physical Function (p = 0.60, panel B) or steps per 24 hours (p = 0.22, panel C). Also shown in Fig 2 are the corresponding analyses for PORH at baseline versus disease severity (p for trend = 0.15, panel D), and the correlations between PORH and SF-36 Physical Function (p = 0.46, panel E). There was a significant correlation between PORH and steps per 24 hours at baseline (r = 0.39, p = 0.014, Fig 2, panel F).

**18 months follow-up, changes in clinical status and vascular measures.** The Bergen cohort of the RituxME trial with rituximab and placebo groups pooled, experienced a slight clinical improvement after 18 months (Table 3).

There were, however, no significant differences in outcome measures between the rituximab and placebo groups, as reported for the complete RituxME trial [11]. Mean values of SF-

**Table 2. Comparison of ME/CFS patients and healthy control groups at baseline.**

| | All patients | | Healthy controls[1] | | Healthy controls[2] | |
|---|---|---|---|---|---|---|
| | N | % or mean (SD/SE) | N | % or mean (SD/SE) | N | % or mean (SD/SE) |
| Age (years) | 39 | 35.9 (SD 9.2) | 30 | 37.2 (SD 11.3) | 66 | 39.1 (SD 5.4)^ |
| Female sex | 39 | 82.1% | 25 | 83.3% | 66 | 100% |
| BMI (kg/m$^2$) | 39 | 23.5 (SD 3.1) | | | 66 | 26.1 (SD 5.2)* |
| Systolic BP (mmHg) | 39 | 123.0 (SD 13.8) | 30 | 115.4 (SD 10.1)^ | 66 | 114.8 (SD 10.9)** |
| Diastolic BP (mmHg) | 39 | 74.6 (SD 10.8) | 30 | 66.3 (SD 9.4)** | 66 | 71.6 (SD 8.6) |
| FMD, baseline (%)[3] | 39 | 5.1 (SE 0.64) | | | 66 | 8.2 (SE 0.48)*** |
| Dilation after Nitro (%)[3,4] | 39 | 23.1 (SE 1.1) | | | 66 | 22.7 (SE 0.83) |
| PORH, baseline (PU)[5] | 39 | 1354 (SD 754) | 30 | 2208 (SD 1363)* | | |

[1]Healthy control group for PORH measurements.

[2]Healthy control group for FMD measurements.

[3]Comparison between ME/CFS patients and healthy control group, adjusted for sex and baseline arterial diameter, by ANCOVA.

[4]Maximal dilation after sublingual nitroglycerin.

[5]Comparison between ME/CFS patients and healthy control group, with Mann-Whitney test.

All p-values indicated are for comparison of ME/CFS patients versus the relevant control group.

^p<0.05;

*p<0.01;

**p = 0.001;

***p<0.0001

Abbreviations: FMD: Flow-mediated dilation; PORH: Post-occlusive reactive hyperemia; PU: perfusion units.

36-PF, Fatigue Severity Scale, self-reported function level in percent, and mean steps per 24 hours were significantly better at 18 months compared to baseline (Table 3). There was a significant increase in PORH from baseline to 18 months, mean PORH at baseline was 1360 PU and at 18 months 1834 PU, p = 0.028 (Table 3). However, FMD values were not similarly improved. Among 36 ME/CFS patients with data at baseline and at 18 months, mean flow-mediated dilation (FMD) at baseline was 5.0% and at 18 months 4.7% (p = 0.68). There was no significant correlation between FMD and PORH at 18 months (r = 0.077, p = 0.66).

Selected characteristics for ME/CFS patients by the randomization groups (placebo and rituximab), at baseline and at 18 months follow-up, are shown in Table 4. Fig 3 shows FMD

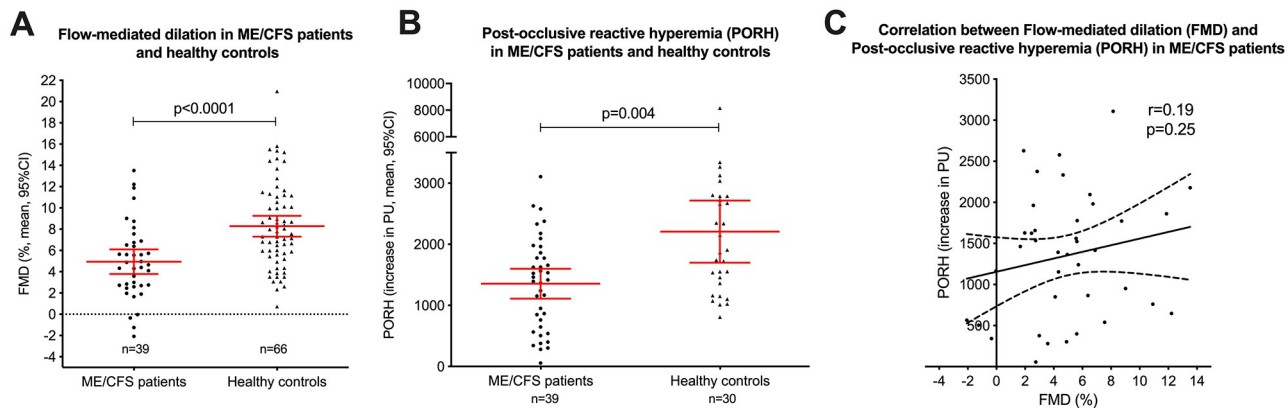

**Fig 1. Flow-mediated dilation and post-occlusive reactive hyperemia in ME/CFS patients and healthy controls.**

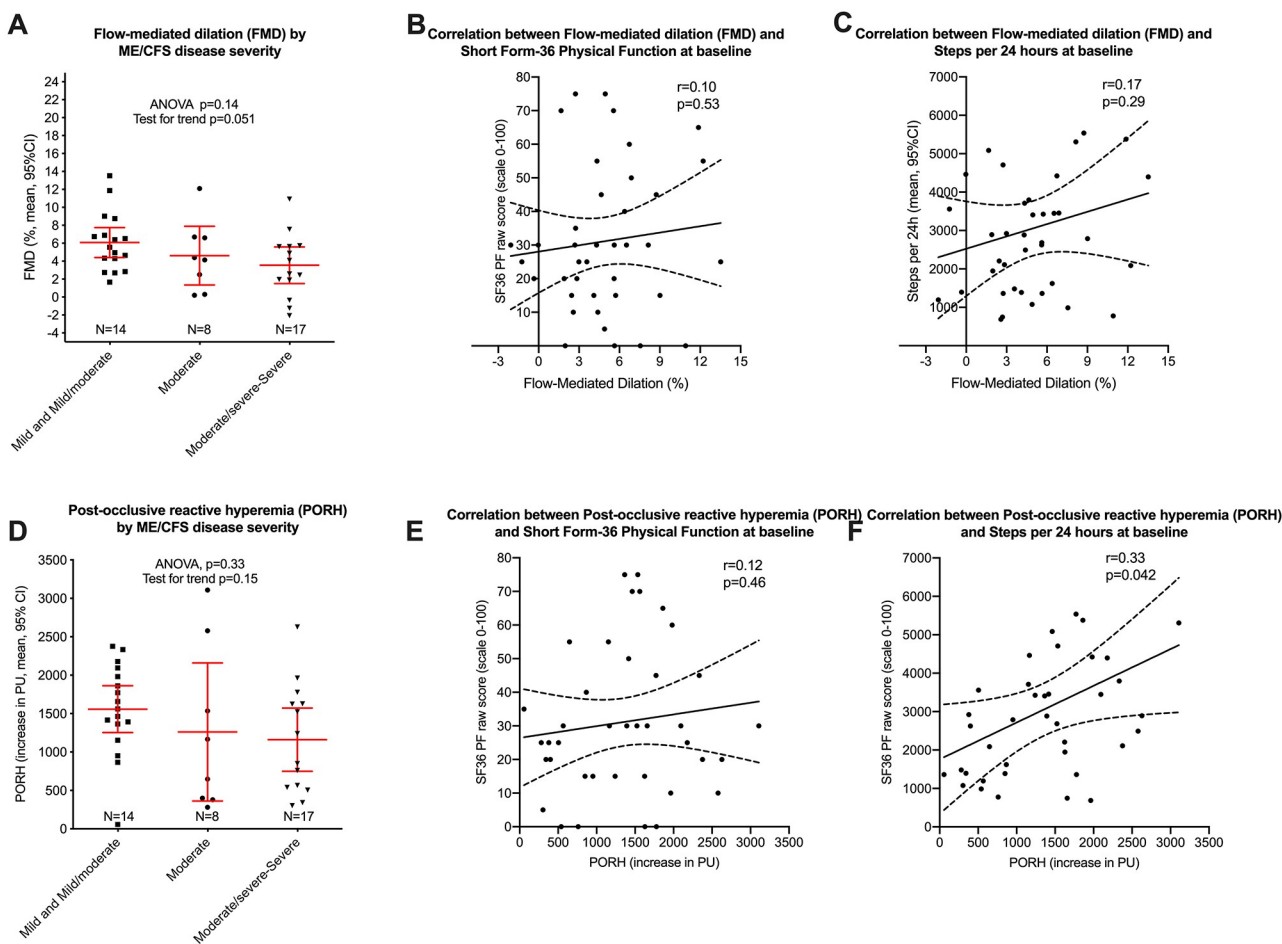

**Fig 2. Associations between vascular measures and disease severity.**

**Table 3. Outcome measures at baseline and at 18 months follow-up.**

| All patients | N | Baseline % or mean (SD) | N | 18 months % or mean (SD) | p-value[5] |
|---|---|---|---|---|---|
| SF-36-PF[1] | 39 | 31.2 (21.7) | 39 | 42.1 (25.8) | 0.007 |
| Fatigue Severity Scale[2] | 39 | 59.7 (3.7) | 39 | 56.0 (10.8) | 0.037 |
| Self-reported function level (%)[3] | 39 | 18.0 (9.1) | 39 | 23.7 (15.5) | 0.022 |
| Mean steps per 24h[4] | 39 | 3022 (2220) | 39 | 3839 (3038) | 0.018 |
| FMD (%) | 36 | 5.0 (3.7) | 36 | 4.7 (3.4) | 0.68 |
| PORH (PU) | 36 | 1360 (725) | 36 | 1834 (967) | 0.028 |

[1]SF-36-PF = Short-Form-36 Health Survey Physical Function subscale: range 0–100, higher scores denote better function.

[2]Fatigue Severity Scale: range 7–63, higher scores denote worse symptoms.

[3]Self-reported function level, range 0–100%, according to a form with examples.

[4]Measured by Sensewear armbands, continuously for 5–7 consecutive days.

[5]p-values from Wilcoxon's test for paired data.

Abbreviations: FMD: Flow-mediated dilation; PORH: Post-occlusive reactive hyperemia; PU: perfusion units.

**Table 4. Characteristics for the rituximab and placebo groups.**

| | *Placebo* | | *Rituximab* | |
|---|---|---|---|---|
| | N | % or mean (SD) | N | % or mean (SD) |
| Age (years) | 20 | 34.0 (9.8) | 20 | 37.9 (8.1) |
| Female sex | 16 | 80.0% | 17 | 85.0% |
| Disease severity (1–5)[1] | 20 | 2.9 (1.4) | 20 | 3.1 (1.5) |
| Disease duration (years)[2] | 20 | 7.9 (2.5) | 20 | 8.0 (3.4) |
| SF-36 Physical Function[3], baseline | 20 | 26.5 (17.1) | 20 | 34.8 (25.3) |
| SF-36 Physical Function, 18 months | 20 | 38.9 (23.0) | 20 | 43.6 (29.1) |
| FMD (%) at baseline | 20 | 5.2% (2.9) | 19 | 4.6% (4.2) |
| FMD (%) at 18 months | 17 | 5.3% (3.4) | 19 | 4.2% (3.4) |
| PORH (PU) at baseline | 17 | 1407 (839) | 19 | 1317 (627) |
| PORH (PU) at 18 months | 17 | 1972 (803) | 19 | 1711 (1100) |

[1]Disease severity in five categories: 1: mild; 2: mild/moderate; 3: moderate; 4 moderate/severe; 5: severe.

[2]Years from diagnosis to inclusion.

[3]SF-36: Short-Form-36 Health Survey Physical Function subscale: range 0–100, higher score denote better function.

Abbreviations: FMD: Flow-mediated dilation; PORH: Post-occlusive reactive hyperemia; PU: perfusion units.

(panel A) and PORH (panel B), at baseline and at 18 months, by randomization group. By GLM repeated measures analyses, there were no significant interactions time-by-randomization group, i.e. there were no significant differences between the rituximab and placebo groups, in the changes of FMD or PORH from baseline to 18 months follow-up (Fig 3). The time effect was significant for PORH (panel B).

## Discussion

The present study is one of the first studies to investigate vascular function in ME/CFS patients, and was a substudy of a multi-center, national, randomized and placebo-controlled

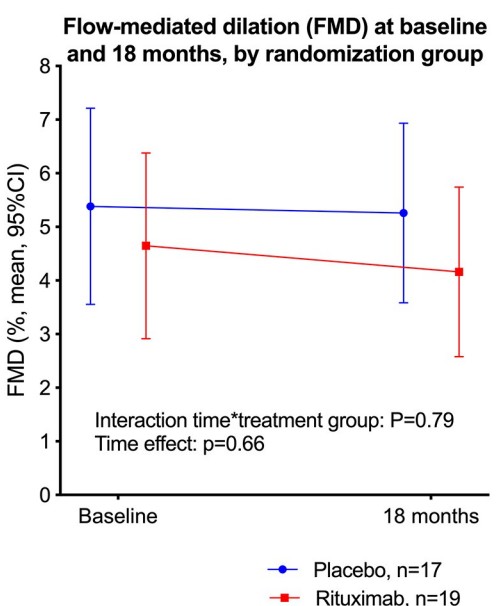

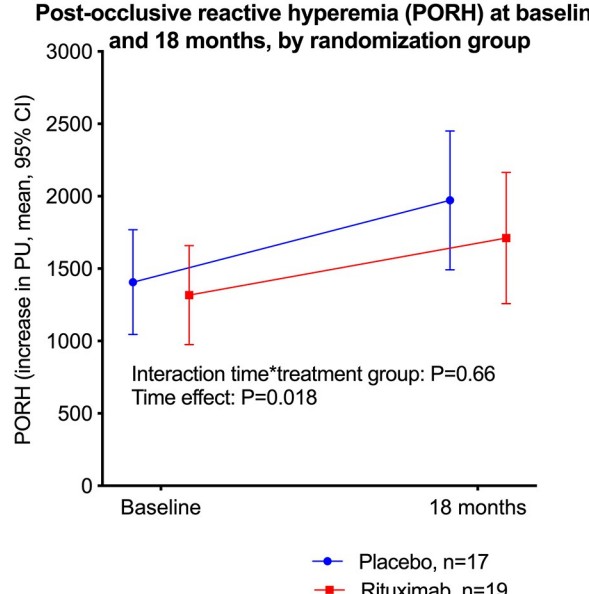

**Fig 3. Vascular measures at baseline and 18 months.**

trial that investigated the effect of B-cell depletion using the monoclonal antibody rituximab in this poorly understood disease. We found significantly lower endothelial function measured with FMD, and lower microvascular regulation measured with PORH, in patients with ME/CFS compared to healthy control groups. The vascular dilation after administration of glyceryl nitrate in ME/CFS patients was similar to healthy controls, showing that the intrinsic ability to dilate adequately was present. We found no significant correlation between FMD and PORH. There were tendencies towards lower FMD and PORH in patients with moderate to severe ME/CFS symptoms. However, there were no significant correlations between measures of vascular function and disease severity or physical function measures except for a significant correlation between PORH and steps per 24 hours at baseline. Overall, the patients experienced a slight clinical improvement during follow-up in the trial. There was no corresponding improvement in endothelial function measured by FMD, but a significant improvement in PORH from baseline to repeated measurement at 18 months. There was no difference in changes of endothelial function (FMD or PORH) during follow-up between the rituximab and placebo groups.

Our findings support our first hypothesis of reduced vascular function in ME/CFS patients, and confirm previous findings [14, 16]. There was also a tendency towards an association between vascular function and disease severity, which lends some support to our second hypothesis. Our third hypothesis was that a normalization of vascular function would occur if the active treatment group with rituximab showed symptom alleviation. Taking into account the negative result from the clinical trial, and thus no support for a beneficial role of rituximab in ME/CFS diagnosed according to Canadian criteria, our results neither support nor disprove this hypothesis.

Nevertheless, it is a significant and important finding that our group of ME/CFS patients had such markedly reduced vascular function compared to healthy individuals. The observed endothelial dysfunction argues for a pathomechanistic effect of the vascular system in ME/CFS.

The mechanisms for vascular dysfunction in ME/CFS patients are unclear. Endothelial dysfunction, in both large and small vessels, is a well-known risk factor for cardiovascular disease later in life [28–30]. Previous studies have indicated that patients with ME/CFS have an increased mortality due to heart failure [31], and the disease is associated with symptoms of cardiovascular stress and risk factors for cardiovascular disease such as autonomic dysfunction [32, 33], impaired blood pressure regulation [34, 35], increased levels of oxidative stress [36], low-grade inflammation, and arterial stiffness [37]. A recent study of invasive cardiopulmonary testing in ME/CFS patients demonstrated impaired peripheral oxygen extraction and reduced cardiac output due to reduced venous return, manifestations of peripheral neurovascular dysregulation that are plausible contributors to ME/CFS exertional intolerance [38]. Our findings of vascular dysfunction lend further support to a hypothesis of endothelial dysfunction in ME/CFS patients. Normal endothelial-independent dilation is a strong indicator of endothelial NO-production being disturbed in these patients. The finding of higher resting blood pressure in ME patients than in the control groups could support such a hypothesis.

We recently summarized our hypotheses on possible pathomechanisms in ME/CFS [39]. We argue for an underlying abnormal immune response, often occurring after infection, involving B-cells and possibly autoantibodies. The immune response may disturb vascular function and cause inadequate autoregulation of blood flow according to the metabolic demands of tissues, resulting in tissue hypoxia with early lactate accumulation on exertion. Tissue hypoxia on exertion could underlie several characteristic symptoms of ME/CFS and may further cause secondary and compensatory mechanisms, such as autonomic and metabolic adaptations.

However, the endothelial dysfunction could also be a secondary phenomenon and therefore not directly associated with disease severity or improvement in patients experiencing symptom fluctuations during follow-up. Another possible mechanism for reduced vascular function could be inactivity. A meta-analysis of the effects of exercise on FMD concluded that exercise training contributed to an increase in FMD [40], indicating that levels of physical activity affect endothelial function. However, Fenton et al reported an inverse relationship between a sedentary lifestyle (sitting time) and microvascular endothelium-dependent function, but not with FMD [41].

In the present study, there was no clear relation between endothelial dysfunction and severity of disease symptoms, although we observed a tendency towards higher FMD and PORH values in patients with milder symptoms.

There was no correlation between microvascular function measured by PORH, and endothelial function of the brachial artery measured by FMD. These methods both measure endothelial-dependent vascular function, but large and small vessels are regulated by somewhat different mechanisms, and studies of both measures have often found a poor or lacking correlation between FMD and PORH [42, 43]. This finding was therefore not unexpected.

Strengths of the present study: The RituxME study was a high quality study, with a double-blind randomized design, and selection of patients according to well-established and stringent criteria. The patients followed a strict and thorough follow-up regime. Endothelial function was measured using two different and complementary methods, both well documented. Vascular measures in the healthy control groups were similar to other studies of healthy individuals in comparable age groups [18, 44]. One of our main findings was that there was no difference in endothelial function at 18 months between the placebo and rituximab groups, and this fits well with the fact that Rituximab had no significant effect on ME/CFS symptoms in this study.

Weaknesses of the present study: Our patient sample was limited (39 patients). However, such patient numbers are not uncommon in FMD studies, and are often sufficient to show significant differences between groups [45, 46]. Another weakness of this study is the FMD method itself. It is a complex and operator-dependent method, and requires diligence and meticulousness in patient preparations and execution of the measurements. Nevertheless, the measurements were done by two well-trained and experienced physicians, with inter-observer variability ranging from good (FMD in percent) to excellent (absolute measures) [47]. For baseline FMD data, we compared the patient group to a previous control group from a different study. This control group was all-female, in contrast to 85% females in the patient group. However, this control group was relatively large, measurements were done by the same operator as most measurements in the present study (MKS), and was comparable in age. The FMD measurements in this control group was also well in line with other FMD data from the general population [18]. The FMD results in our study were similar when adjusted for sex, and when men were excluded from the patient group.

Clinical implications of this study: We think our findings of reduced endothelial function, both micro- and macrovascular, are of clinical importance. FMD and PORH measurements are not regarded as useful diagnostic tools for any patients groups, but are used to evaluate risk on a group level. Our study supports growing evidence that ME/CFS patients have a markedly reduced endothelial function compared to the normal population. This is an important discovery in a field with a scarcity of objective and measurable abnormalities. Our study does not answer the question of why these patients have reduced endothelial function. Reduced endothelial function might be an associated symptom rather than a central mechanism of the disease. However, we suspect that in ME/CFS the endothelial dysfunction is related to an abnormal immune response, and may be a factor in the pathomechanism. Together with

reduced venous tone ("preload failure"), and arteriovenous anastomoses [38], arterial endothelial dysfunction may disturb the autoregulation of blood flow to meet the metabolic demands of tissues, resulting in tissue hypoxia on exertion [39]. These findings reveal a great need for further studies on endothelial function and possible disease mechanisms in these patients. Furthermore, the reduced endothelial function and higher blood pressure compared to the healthy control group might indicate a long-term increased risk of developing cardiovascular disease for ME/CFS patients.

## Conclusions

ME/CFS patients in this study had reduced endothelial function, indicating that vascular homeostasis is affected in at least some of these patients and may play a role in the clinical presentation of this disease. There is a pressing need for more studies to investigate this important issue, in order to better understand causes and disease mechanisms.

## Supporting information

**S1 File. Study protocol.**
(PDF)

**S2 File. GLM syntax.**
(PDF)

**S3 File. Study file.** Patients and PORH controls.
(XLSX)

**S4 File. Study file.** Patients and FMD controls.
(XLSX)

## Author Contributions

**Conceptualization:** Miriam Kristine Sandvik, Kari Sørland, Elisabeth Leirgul, Ingrid Gurvin Rekeland, Olav Mella, Øystein Fluge.

**Data curation:** Miriam Kristine Sandvik, Kari Sørland, Elisabeth Leirgul, Christina Særsten Stavland, Øystein Fluge.

**Formal analysis:** Miriam Kristine Sandvik, Elisabeth Leirgul, Øystein Fluge.

**Funding acquisition:** Olav Mella, Øystein Fluge.

**Investigation:** Miriam Kristine Sandvik, Kari Sørland, Elisabeth Leirgul, Christina Særsten Stavland, Øystein Fluge.

**Methodology:** Miriam Kristine Sandvik, Kari Sørland, Elisabeth Leirgul, Ingrid Gurvin Rekeland, Olav Mella, Øystein Fluge.

**Project administration:** Miriam Kristine Sandvik, Kari Sørland, Elisabeth Leirgul, Ingrid Gurvin Rekeland, Olav Mella, Øystein Fluge.

**Resources:** Elisabeth Leirgul, Olav Mella, Øystein Fluge.

**Supervision:** Kari Sørland, Elisabeth Leirgul, Olav Mella, Øystein Fluge.

**Validation:** Miriam Kristine Sandvik, Kari Sørland, Elisabeth Leirgul, Olav Mella, Øystein Fluge.

**Visualization:** Miriam Kristine Sandvik, Ingrid Gurvin Rekeland, Olav Mella, Øystein Fluge.

**Writing – original draft:** Miriam Kristine Sandvik, Kari Sørland, Elisabeth Leirgul, Ingrid Gurvin Rekeland, Christina Særsten Stavland, Olav Mella, Øystein Fluge.

**Writing – review & editing:** Miriam Kristine Sandvik, Kari Sørland, Elisabeth Leirgul, Ingrid Gurvin Rekeland, Christina Særsten Stavland, Olav Mella, Øystein Fluge.

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
