## [Decision Letter · Decision Letter 0]

15 Sep 2022

PONE-D-22-19007Endothelial dysfunction in ME/CFS patients.PLOS ONE

Dear Dr. Sandvik,

Thank you for submitting your manuscript to PLOS ONE. After careful consideration, we feel that it has merit but does not fully meet PLOS ONE’s publication criteria as it currently stands. Therefore, we invite you to submit a revised version of the manuscript that addresses the points raised during the review process.

The reviewer report can be found at the end of this email. Please note that we have only been able to secure a single reviewer to assess your manuscript. We are issuing a decision on your manuscript at this point to prevent further delays in the evaluation of your manuscript. Please be aware that the editor who handles your revised manuscript might find it necessary to invite additional reviewers to assess this work once the revised manuscript is submitted. However, we will aim to proceed on the basis of this single review if possible. 

We look forward to receiving your revised manuscript.

Kind regards,

Debora Walker

Editorial Office

PLOS ONE

Journal Requirements:

6. We note that you have included the phrase “data not published” in your manuscript. Unfortunately, this does not meet our data sharing requirements. PLOS does not permit references to inaccessible data. We require that authors provide all relevant data within the paper, Supporting Information files, or in an acceptable, public repository. Please add a citation to support this phrase or upload the data that corresponds with these findings to a stable repository (such as Figshare or Dryad) and provide and URLs, DOIs, or accession numbers that may be used to access these data. Or, if the data are not a core part of the research being presented in your study, we ask that you remove the phrase that refers to these data.

7. Please include a separate caption for each figure in your manuscript.

Reviewers' comments:

Reviewer's Responses to Questions

**Comments to the Author**

1. Is the manuscript technically sound, and do the data support the conclusions?

Reviewer #1: No

2. Has the statistical analysis been performed appropriately and rigorously? 

Reviewer #1: No

3. Have the authors made all data underlying the findings in their manuscript fully available?

Reviewer #1: No

4. Is the manuscript presented in an intelligible fashion and written in standard English?

Reviewer #1: Yes

5. Review Comments to the Author

Reviewer #1: PONE-D-22-19007: statistical review

SUMMARY. This is a study of vessel endothelial function in patients with Myalgic Encephalomyelitis/Chronic Fatigue Syndrome (ME/CFS). Two main outcomes (flow-mediated dilation, FMD, and post-occlusive reactive hyperemia, PORH) and supplementary measures of symptom severity and physical functioning have been measured at baseline and after 18 months of treatment in 39 patients and compared with healthy controls. The study seems well designed and the research questions are clearly stated. However, there are some important points that need further clarification: see the specific issues below.

SPECIFIC ISSUES.

1. Lines 136-138 say that the severity of ME/CFS has been categorized according to self-reported function level and the physicians’ evaluation at the time of inclusion. However, nothing is said about how these different sources of information were combined. The authors should provide a detailed description of the protocol used to define the severity classes.

2. Lines 202-204 say that some variables were not normally distributed, without specifying which variables. This creates some confusion. For example, some variables are first analyzed by ANCOVA methods (Table 2), which require normality, and then examined by the the Wilcoxon’s test (Table 3), as they were non-normal. The authors should specify which variables are normally distributed and the test used to check normality.

3. Lines 205-208 say that general linear models (GLM) for repeated measures were used to assess differences in changes of the outcomes measures (FMD or PORH), from baseline to 18 months, between the randomization groups (rituximab and placebo). Although this approach is in principle correct, I was not able to understand the structure of the model, because the outcome of this analysis is displayed only by a battery of pictures. I'd welcome a traditional table with the estimates (and p-values) of the coefficients of the model, including an estimate of the heterogeneity parameter (the variance of the random effects).

6. PLOS authors have the option to publish the peer review history of their article (what does this mean?). If published, this will include your full peer review and any attached files.

Reviewer #1: No

---

## [Author Response · Author response to Decision Letter 0]

31 Oct 2022

Response to reviewers regarding the article "Endothelial dysfunction in ME/CFS patients" 

(PONE-D-22-19007)

We want to thank the reviewer and the editors for the positive evaluation of our manuscript, and for their valuable comments. We will respond to the comments made by the reviewer and the additional comments by the editors point by point, as requested (all references to manuscript text refer to the version with marked changes):

Reviewer #1: 

SPECIFIC ISSUES.

1. Lines 136-138 say that the severity of ME/CFS has been categorized according to self-reported function level and the physicians’ evaluation at the time of inclusion. However, nothing is said about how these different sources of information were combined. The authors should provide a detailed description of the protocol used to define the severity classes.

Answer: We agree that our description of the severity categories are insufficient. We have added this text: (lines 142-145): “The categorization into different severity groups was based on thorough clinical assessment by the physician at the time of inclusion, and supported by patient-reported function level and questionnaires.»

The following list was used as a guideline for the clinicians (originally in Norwegian, translated here): 

• Severe ME/CFS often has a 5-10% function level. 

• Moderate/severe ME/CFS often has an 8-10-13% function level.

• Moderate ME/CFS often has a 12-15-18% function level. 

• Mild/moderate ME/CFS often has an 18-25% function level.

• Mild ME/CFS often has a 25-40% function level.

• Very severe ME/CFS is typically < 5%, and should not be included in the RituxME study

We also submit the trial protocol as supplementary material (S1 File). 

2. Lines 202-204 say that some variables were not normally distributed, without specifying which variables. This creates some confusion. For example, some variables are first analyzed by ANCOVA methods (Table 2), which require normality, and then examined by the Wilcoxon’s test (Table 3), as they were non-normal. The authors should specify which variables are normally distributed and the test used to check normality.

Answer: We agree with the reviewer that the description of our choice of analyses were unclear. All variables have been tested for normality using the Shapiro-Wilk´s test. For the vascular variables, FMD was normally distributed in both patients and controls, while PORH was normally distributed in patients, but not in the control group. The patient variables Fatigue Severity Scale, self-reported function level, SF-36 Physical Function and Steps per 24 h were not normally distributed. 

We have specified and explained our choices of statistical tests more thoroughly in the "Statistical analyses" section (lines 207-223 and 229-233) and in the tables, and hope this is satisfactory. We have, also, reanalyzed the correlation analyses including PORH measurements with non-parametric tests, and made corresponding changes in the "Results" section (lines 295, 297 and 309-315).

The figures have been adjusted with p-values from the appropriate analyses, and with error bars corresponding to the type of analyses (mean, SD for normally distributed data, and median, IQR for data not normally distributed). 

3. Lines 205-208 say that general linear models (GLM) for repeated measures were used to assess differences in changes of the outcomes measures (FMD or PORH), from baseline to 18 months, between the randomization groups (rituximab and placebo). Although this approach is in principle correct, I was not able to understand the structure of the model, because the outcome of this analysis is displayed only by a battery of pictures. I'd welcome a traditional table with the estimates (and p-values) of the coefficients of the model, including an estimate of the heterogeneity parameter (the variance of the random effects).

Answer: We have included the analysis syntax for the GLM repeated measures as a supporting file (S2 File), which should give the details of the model. We also added in the Figure 3 the differences with confidence intervals between groups, for FMD og PORH. 

Additional requirements by the editors:

Answer: We have made changes in the manuscript to meet PLOS ONE´s style requirements. These changes are not marked in the manuscript with marked changes. 

Answer: We have added a section on participant consent in the Methods section (lines 127-132). 

Answer: We thank the editors for pointing this out – the discrepancies may be due to different translations of funding organizations. We have corrected the information in the "Funding information" section at resubmission. 

Answer: After re-evaluation, we find that it is acceptable to make an anonymized datafile available, and have submitted this as supplementary information (S3 File for patients and PORH controls, S4 File for patients and FMD controls). 

Answer: We refer to our answer above regarding data availability. 

6. We note that you have included the phrase “data not published” in your manuscript. Unfortunately, this does not meet our data sharing requirements. PLOS does not permit references to inaccessible data. We require that authors provide all relevant data within the paper, Supporting Information files, or in an acceptable, public repository. Please add a citation to support this phrase or upload the data that corresponds with these findings to a stable repository (such as Figshare or Dryad) and provide and URLs, DOIs, or accession numbers that may be used to access these data. Or, if the data are not a core part of the research being presented in your study, we ask that you remove the phrase that refers to these data.

Answer: We have deleted two references to unpublished data/observations in lines 82-85 and lines 438-440. We have kept a brief description of exploratory FMD measures from 2014 ( lines 85-91) because these findings were relevant in the planning of the substudy described in this paper. We do not have the participants´ consent to make these raw data available in a supplementary file. If the editors find this unsatisfactory, we suggest to delete all mention of these exploratory measures (lines 85-91).

7. Please include a separate caption for each figure in your manuscript.

Answer: We have included a caption for each figure as requested. 

In addition to the changes made upon request, we suggest to delete correlations between vascular measures and disease severity at 18 months, lines 316-319. We also made some minor changes in wording and grammar throughout the text, those changes are marked in the manuscript with marked changes, but not referenced in detail here.

---

## [Decision Letter · Decision Letter 1]

12 Jan 2023

Endothelial dysfunction in ME/CFS patients.

PONE-D-22-19007R1

Dear Dr. Sandvik,

We’re pleased to inform you that your manuscript has been judged scientifically suitable for publication and will be formally accepted for publication once it meets all outstanding technical requirements.

Kind regards,

Doan TM Ngo, B.Pharm., PhD.

Academic Editor

PLOS ONE

Additional Editor Comments (optional):

Reviewers' comments:

Reviewer's Responses to Questions

**Comments to the Author**

1. If the authors have adequately addressed your comments raised in a previous round of review and you feel that this manuscript is now acceptable for publication, you may indicate that here to bypass the “Comments to the Author” section, enter your conflict of interest statement in the “Confidential to Editor” section, and submit your "Accept" recommendation.

Reviewer #1: All comments have been addressed

2. Is the manuscript technically sound, and do the data support the conclusions?

Reviewer #1: (No Response)

3. Has the statistical analysis been performed appropriately and rigorously? 

Reviewer #1: (No Response)

4. Have the authors made all data underlying the findings in their manuscript fully available?

Reviewer #1: (No Response)

5. Is the manuscript presented in an intelligible fashion and written in standard English?

Reviewer #1: (No Response)

6. Review Comments to the Author

Reviewer #1: (No Response)

7. PLOS authors have the option to publish the peer review history of their article (what does this mean?). If published, this will include your full peer review and any attached files.

Reviewer #1: No

---

## [Editor Report · Acceptance letter]

25 Jan 2023

PONE-D-22-19007R1 

Endothelial dysfunction in ME/CFS patients 

Dear Dr. Sandvik:

I'm pleased to inform you that your manuscript has been deemed suitable for publication in PLOS ONE. Congratulations! Your manuscript is now with our production department. 

Kind regards, 

on behalf of

Dr. Doan TM Ngo 

Academic Editor

PLOS ONE